# Tracing the path of 37,050 studies into practice across 18 specialties of the 2.4 million published between 2011 and 2020

**Moustafa Abdalla[1,2]\*, Salwa Abdalla[3], Mohamed Abdalla[3,4]**

[1]Department of Surgery, Harvard Medical School, Massachusetts General Hospital, Boston, Canada; [2]Computational Statistics and Machine Learning Group, Department of Statistics, University of Oxford, Oxford, United Kingdom; [3]Department of Computer Science, University of Toronto, Toronto, Canada; [4]Institute for Better Health, Trillium Health Partners, Ontario, Canada

**\*For correspondence:**
moustafa_abdalla@hms.harvard.edu

**Competing interest:** The authors declare that no competing interests exist.

**Abstract** The absence of evidence to assess treatment efficacy partially underpins the unsustainable expenditure of the US healthcare system, a challenge exacerbated by a limited understanding of the factors influencing the translation of clinical research into practice. Leveraging a dataset of >10,000 UpToDate articles, sampled every 3 months between 2011 and 2020, we trace the path of research (37,050 newly added articles from 887 journals) from initial publication to the point-of-care, compared to the 2.4 million uncited studies published during the same time window across 18 medical specialties. Our analysis reveals substantial variation in how specialties prioritize/adopt research, with regards to a fraction of literature cited (0.4–2.4%) and quality-of-evidence incorporated. In 9 of 18 specialties, less than 1 in 10 clinical trials are ever cited. Furthermore, case reports represent one of the most cited article types in 12 of 18 specialties, comprising nearly a third of newly added references for some specialties (e.g. dermatology). Anesthesiology, cardiology, critical care, geriatrics, internal medicine, and oncology tended to favor higher-quality evidence. By modeling citations as a function of National Institutes of Health (NIH) department-specific funding, we estimate the cost of bringing one new clinical citation to the point-of-care as ranging from thousands to tens of thousands of dollars depending on specialty. The success of a subset of specialties in incorporating a larger proportion of published research, as well as high(er) quality of evidence, demonstrates the existence of translational strategies that should be applied more broadly. In addition to providing a baseline for monitoring the efficiency of research investments, we also describe new 'impact' indices to assess the efficacy of reforms to the clinical scientific enterprise.

## Editor's evaluation

This paper presents an analysis of how clinical research makes its way into clinical practice in different medical specialties and across time. This is an important analysis since it has the potential to identify bottlenecks in the pathway from research to practice. The strength of evidence is convincing in that a substantial number of publications are considered over a wide time frame.

## Introduction

Two key elements to the underperformance of national healthcare systems are that: (a) many patients do not receive recommended services (*Gelijns and Gabriel, 2012*; *McGlynn et al., 2003*; *MEDICINE IROE-B, 2011*) and (b) many receive treatment that is neither necessary nor appropriate for them (*McGlynn et al., 2003*; *MEDICINE IROE-B, 2011*; *Fisher and Wennberg, 2003*). The Institute

of Medicine (IOM) Roundtable on Value and Science-Driven Healthcare argues; however, the challenge is not a matter of overuse (or underuse) of services, but the absence of evidence to assess the appropriateness of treatment approaches (*MEDICINE IROE-B, 2011*; *Wennberg et al., 2002*; *Gill et al., 1996*; *Lee et al., 2005*; *Lee et al., 2000*). With more than 1 million medical research articles published in the past year alone, the adoption of clinical studies into practice is one critical aspect of this challenge – further compounded by a limited understanding of how the wave of biomedical literature reaches the shores of clinical practice. A few case studies/series have attempted to understand this block in clinical adoption using surrogate markers, such as submission to the Food and Drug Administration (FDA) (*Stern and Simes, 1997*; *Holgate, 2007*), number of citations (*Dickersin, 1990*; *Mansfield, 1991*), or incorporation into society-specific clinical guidelines (*Contopoulos-Ioannidis et al., 2008*; *Decullier et al., 2005*; *Buxton et al., 2008*). However, these studies are often too coarse and indirect for a real-time and practical understanding of how clinicians read, synthesize, and integrate the literature into their everyday practice. Furthermore, these studies often conflate translation of basic science with translation of clinical studies to practice, which the IOM has identified as two separate and distinct translational blocks (*Green et al., 2003*). In addition, using citation in consensus documents or society recommendations is too slow and often limited in scope to provide answers to the questions defined here.

Focusing on the translation of clinical studies into practice, we capitalize on the electronic resource UpToDate, which provides current evidence-based clinical information at the point-of-care and is used by over a million clinicians across 32,000 organizations in 180 countries (*Post, 2020*). While the relevance of UpToDate varies, it serves as a reliable and regularly updated source of a specialty-focused clinician-driven curation of the broader literature. Thus, we use citation in UpToDate as one metric to assess translation, especially given its quantifiable impact on patient care (*Shimizu et al., 2018*; *Maviglia et al., 2002*; *Lucas et al., 2004*; *Marshall et al., 2013*; *Bonis et al., 2008*). Leveraging a dataset of more than 10,000 UpToDate articles, sampled every 3 months for the past decade (2011–2020), we provide the first thorough and most comprehensive characterization and understanding of the factors that influence the adoption of clinical research by tracing the path of 37,050 newly added references from 887 journals, as well as provide valuable insight into the variation of adoption across 18 non-surgical specialties by clinical topic, article type, geography, and over time.

## Results

### What fraction of the published literature is eventually cited in point-of-care resources?

Among the 18 specialties included in our analysis, neurology had the highest citation rate; of the 85,843 research articles published in clinical neurology journals during our sampling window, 2057 (2.4%) were eventually cited at least once in UpToDate. Rheumatology (1442 cited of 62,681 published; 2.3%), hematology (2506 of 110,055; 2.3%), and pediatrics (2678 of 119,486; 2.2%) had similar citation rates. Three specialties had sub-percent citation rates: radiology (1214 cited of 165,985 published; 0.7%), geriatrics (64 of 9781; 0.6%), and pathology (317 of 69,343; 0.4%). All remaining specialties, including internal medicine, had between 1 and 2% of all published research eventually cited in UpToDate.

The proportion of citations also varied substantially by article type (*Figure 1*). Practice guidelines represented the most likely article type to be cited, with 9 of the 18 specialties citing >13% (interquartile range [IQR] of 5.1–14.5%) of all practice guidelines published in their respective journals. Although clinical trials (especially phase III trials) were the second most likely article type to be cited (9 of 18 specialties citing >9.5% of all phase III clinical trials published during our sampling window [IQR 3.0–13.0%]), it was also the most variable (SD of 8.7%). In 9 of the 18 specialties, we observed that less than 1 in 10 phase III clinical trials were ever cited at the point-of-care (*Figure 1*). Of the top-performing specialties, the citation rate of clinical trials was distinctly high in internal medicine (299 cited of 822 phase III clinical trials published; 36.3%), pediatrics (8 of 48; 16.7%), and infectious diseases (15 of 99; 15.1%). Notably, no equivalence trial, among the 43 published across all 18 specialties, was ever cited. Comparatively, pragmatic clinical trials were only cited in 5 of the 18 specialties: oncology (50% of published pragmatic clinical trials cited), internal medicine (20.3%), endocrinology (11.1%), cardiology (8.0%), and pediatrics (7.7%). The remaining 13 specialties had a 0% citation rate for pragmatic clinical trials. Similarly, case reports were also unlikely to be cited at the point-of-care

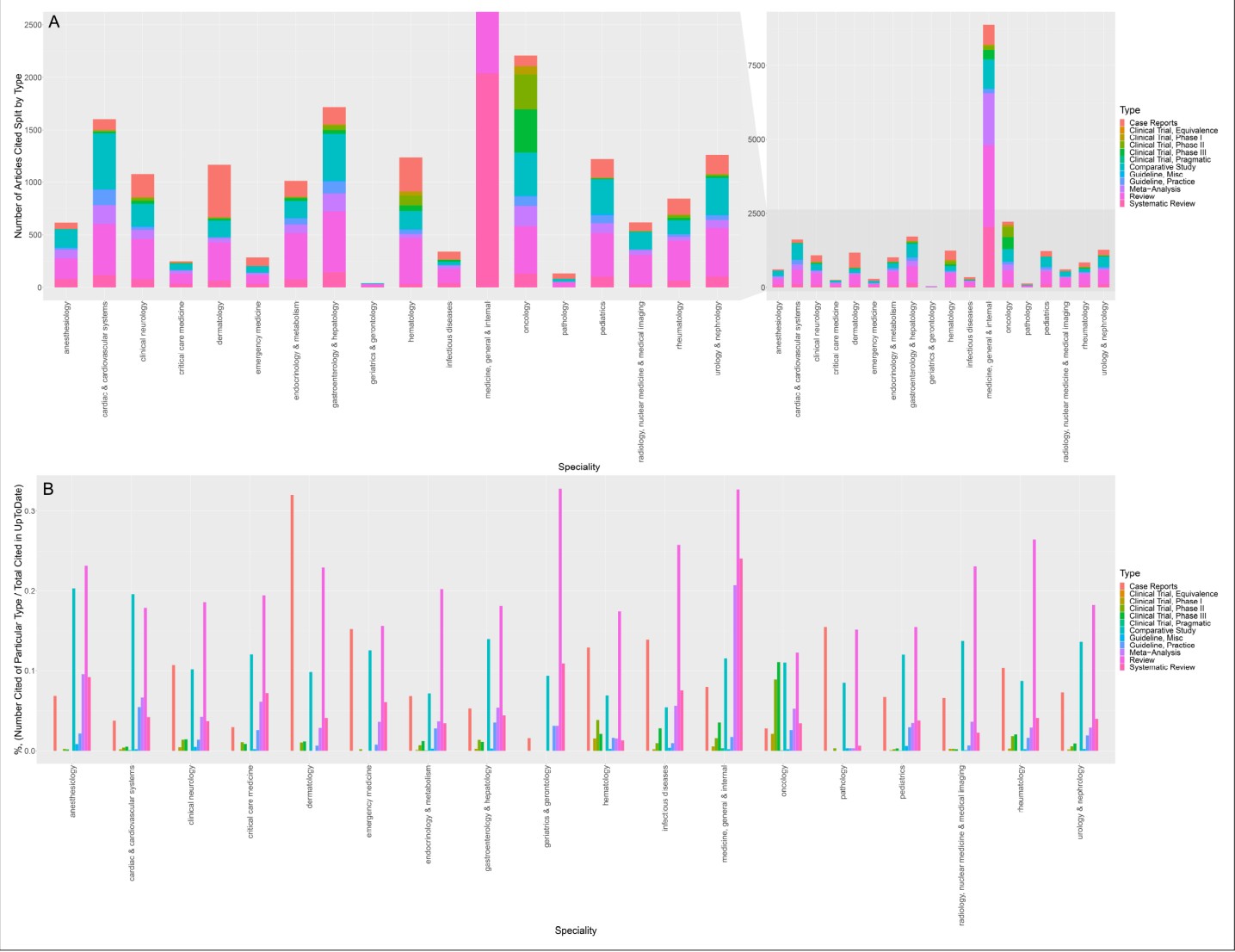

**Figure 1.** Citations in UpToDate over the past decade as a function of article type, grouped by specialty. (**A**) Stacked barplot of the absolute number of citations in UpToDate over the past decade as a function of article type, grouped by specialty (inset: zoomed plot to permit comparison of non-medicine specialties). (**B**) Barplot of the % proportion of articles cited (defined as number of particular article type cited/total number of articles cited in UpToDate) ×100, grouped by medical specialty. (NB:) Totals in panel A may differ slightly from the totals described in the text as some article may have more than one classification and some references are unclassified.

The online version of this article includes the following figure supplement(s) for figure 1:

**Figure supplement 1.** Boxplot of time-to-citation in UpToDate from first publication date as a function of article type, grouped by medical specialty.

across all specialties, with only 3111 case reports (0.8%) cited of the 403,043 published in specialty journals during our sampling window.

## Which are the predominant article types cited in point-of-care resources?

Despite a cumulative citation rate of <1%, case reports still represented the most common article type of those cited in 2 of the 18 specialties (*Figure 1*). Among the 1506 citations added from dermatology journals over the past decade, 501 (32.0%) were case reports. Similarly, of the 317 citations from pathology journals, 49 (15.5%) were case reports. Strikingly, case reports were also consistently among the three most commonly cited article types across all but six specialties (median of 7.1% of added citations were case reports; IQR 5.6–12.3%). By comparison, phase III

clinical trials represented less than 1.0% of added citations in 9 of 18 specialties (IQR 0.2–1.9%). Of the 18 specialties, anesthesiology, cardiac and cardiovascular systems, critical care, geriatrics, internal medicine, and oncology tended to favor higher-quality evidence (*Figure 1*); reviews/systematic reviews, practice-guidelines, and meta-analyses represented the three most cited article types among five of these six specialties. Oncology was relatively unique in that it was the only specialty where phase III clinical trials ranked among the most commonly cited article types; we counted 411 phase III clinical trials among the 3071 references added during our sampling window from oncology journals.

### What is the time-to-citation by specialty and article type?

Time-to-citation did not vary meaningfully between specialties; 50% of articles were cited within a year of publication (IQR 0–4 years). There were significant differences, however, between article types (*Figure 1—figure supplement 1*). Phase III clinical trials had the shortest time-to-citation, with 75% cited within the year of publication (IQR 0–1 year). Meta-analyses, practice guidelines, and systematic reviews followed a similar, albeit slightly slower, trend. Case reports had the longest time-to-citation (median 3 years; IQR 1–9 years). Across all specialties, higher quality of evidence correlated with a shorter time-to-citation (*Figure 1—figure supplement 1*).

### Is journal impact factor predictive of either proportion of articles cited or time-to-citation in point-of-care resources?

For 12 of the 18 medical specialties, journal impact factor was significantly correlated with the proportion of articles cited (*Figure 2*). In descending order, impact factor was significantly correlated with citation rate in: rheumatology (Spearman's rho = 0.86, $p=1.4 \times 10^{-6}$), infectious diseases (rho = 0.79, $p=7.7 \times 10^{-5}$), hematology (rho = 0.69, $p=8.1 \times 10^{-5}$), pediatrics (rho = 0.66, p=0.0001), gastroenterology and hepatology (rho = 0.53, $p=3.6 \times 10^{-5}$), cardiac and cardiovascular systems (rho = 0.55, $p=1.2 \times 10^{-6}$), internal medicine (rho = 0.49, $p=2.3 \times 10^{-6}$), neurology (rho = 0.43, p=0.0086), dermatology (rho = 0.39, p=0.02), urology and nephrology (rho = 0.37, p=0.007), endocrinology and metabolism (rho = 0.32, p=0.02), and oncology (rho = 0.29, p=0.01). In other words, in these 12 specialties, journals with higher impact factors tended to have a larger fraction of their published articles cited at the point-of-care. *Figure 2* visualizes the respective scatterplots labeled by journal. For the remaining six specialties, the relationship between impact factor and portion of cited articles was not significant (p>0.05).

Analogously, journal impact factor was significantly and negatively correlated with time-to-citation for 10 of 18 specialties: infectious diseases (Spearman's rho = –0.51, p=0.03), internal medicine (rho = –0.408, p=0.0001), hematology (rho = –0.407, p=0.03), pediatrics (rho = –0.40, p=0.03), dermatology (rho = –0.406, p=0.02), pathology (rho = –0.45, p=0.04), neurology (Spearman's rho = –0.37, p=0.03), urology and nephrology (rho = –0.34, p=0.02), cardiac and cardiovascular systems (rho = –0.36, p=0.002), and oncology (rho = –0.31, p=0.006). In other words, articles from higher impact specialty journals tended to have a quicker time to citation (*Figure 2—figure supplement 1*).

While the impact factor appears able to (partially) prioritize journals with greater (or quicker) than expected contributions to clinical practice, we sought to better quantify the impact of the journal on clinical practice using two previously introduced new indices (see Materials and methods): the clinical relevancy index (CRI) and the clinical immediacy index (CII). We calculated these indices for all journals in the 18 medical specialties discussed here in *Supplementary file 1*.

### What topics are over-(or under-)represented in abstracts cited in point-of-care resources, compared to uncited literature?

Do these topics explain variation in time-to-citation?

Abstract contents (as assessed using Unified Medical Language System [UMLS] concepts or terms) can significantly explain citation (versus not) in point-of-care resources, as well as variation in time-to-citation among cited abstracts (*Figure 3* and *Figure 3—figure supplements 1–4*). While results for all 18 specialties are fascinating and informative, the Appendix 1 focuses on two specialties (cardiac and cardiovascular systems and endocrinology and metabolism).

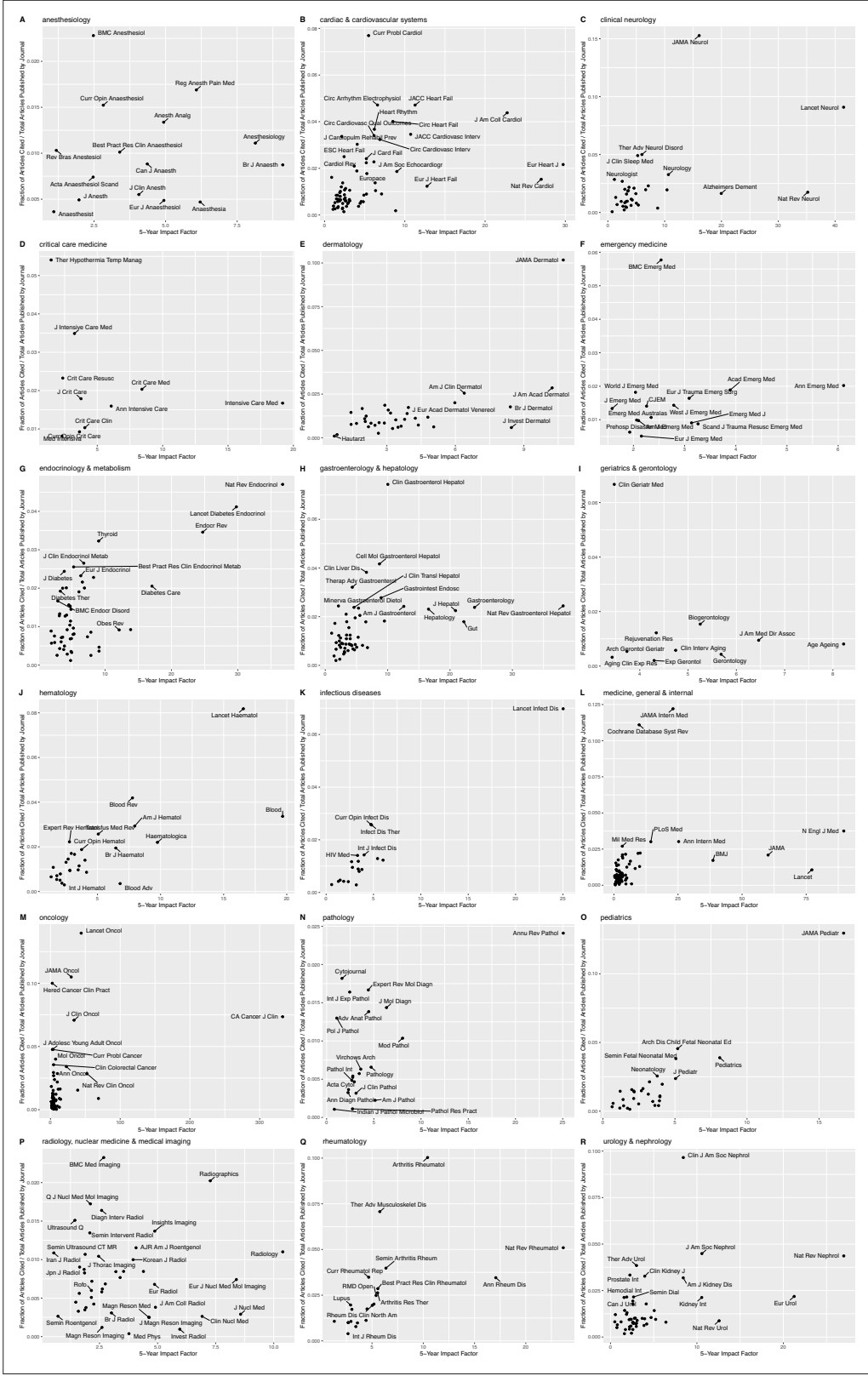

**Figure 2.** Proportion of articles cited as a function of the journal's 5 year impact factor. Scatterplot of proportion of articles cited, defined as the number of citations/total number of articles published by a specific journal, as a function of the journal's 5 year impact factor for (**A**) anesthesiology, (**B**) cardiac and cardiovascular systems, (**C**) clinical neurology, (**D**) critical care medicine, (**E**) dermatology, (**F**) emergency medicine, (**G**) endocrinology and

*Figure 2 continued on next page*

*Figure 2 continued*

metabolism, (**H**) gastroenterology and hepatology, (**I**) geriatrics and gerontology, (**J**) hematology, (**K**) infectious diseases, (**L**) medicine (general and internal), (**M**) oncology, (**N**) pathology, (**O**) pediatrics, (**P**) 'radiology, nuclear medicine and medical imaging,' (**Q**) rheumatology, and (**R**) urology and nephrology.

The online version of this article includes the following figure supplement(s) for figure 2:

**Figure supplement 1.** Median time-to-citation as a function of the journal's 5 year impact factor.

## What is the influence of department-specific NIH funding on the absolute number of citations and time-to-citation?

### What is the impact of cumulative NIH funding?

As we previously noted in curation of our dataset, it is often difficult to disentangle hospitals, medical schools, and affiliated research institutions. As such, to explore the role and impact of NIH funding, we use city (rather than individual institutions or hospitals) as the unit of analysis. Our analysis primarily focused on the United States for two reasons. Firstly, 35% of references from the 18 medical specialties cited in UpToDate were from the United States; by way of comparison, in 2019, 39% of all publications in PubMed were from the United States. Thus, our data was well powered for our funding analyses. Secondly, the NIH publicly releases its funding information with sufficient granularity and standardized specialty labels to enable the analysis.

Our department-specific analysis combined eight specialties (cardiac and cardiovascular systems, critical care medicine, endocrinology and metabolism, gastroenterology and hepatology, geriatrics

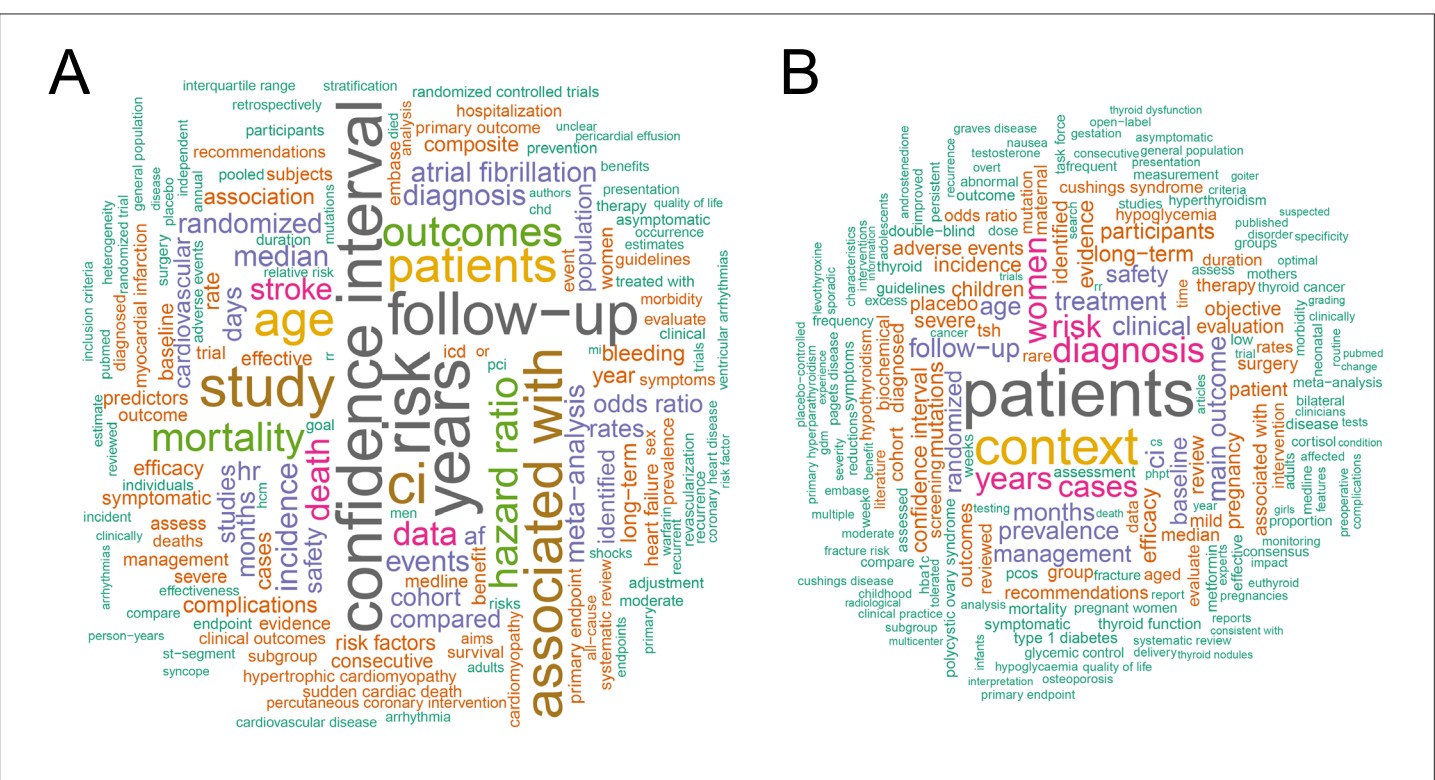

**Figure 3.** Word cloud plots of overrepresented words. Word cloud plots with size of word proportional to the strength of the association for (**A**) Unified Medical Language System (UMLS) concepts significantly overrepresented in articles cited from cardiac and cardiovascular systems (i.e. 'cardiology' journals); (**B**) UMLS concepts significantly overrepresented in articles cited from endocrinology and metabolism journals.

The online version of this article includes the following figure supplement(s) for figure 3:

**Figure supplement 1.** Word cloud plots of overrepresented terms.

**Figure supplement 2.** Word cloud plots of significantly underrepresented terms.

**Figure supplement 3.** Word cloud plots of terms associated a shorter time-to-citation.

**Figure supplement 4.** Word cloud plots of terms significantly associated a longer time-to-citation.

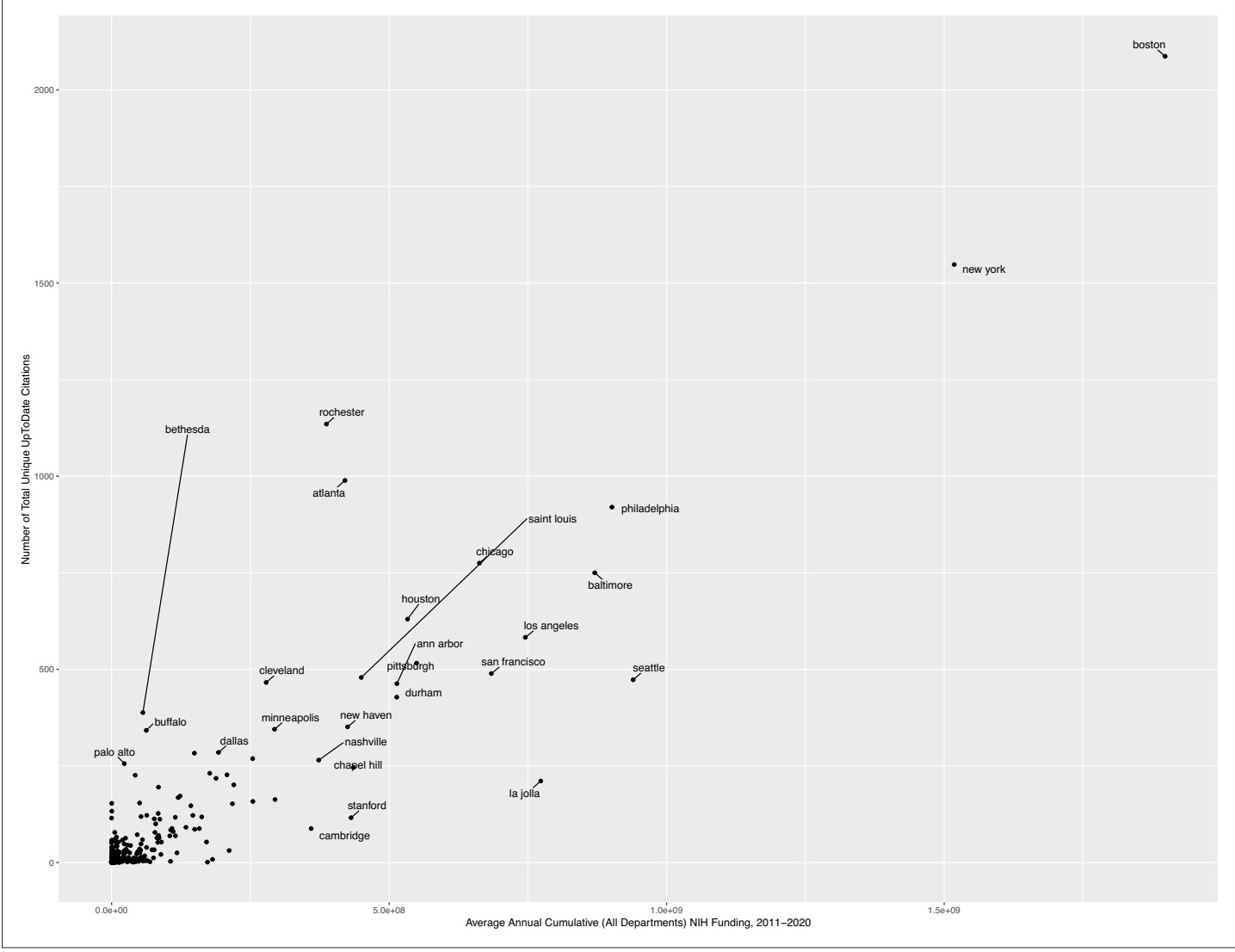

**Figure 4.** Scatterplot of absolute number of articles cited, per city, as a function of the average 10 year cumulative NIH funding (i.e. all citations in all journals across all specialties). It is important to note that this includes all NIH departments (including all medical and surgical specialties, as well as more translational/basic funding departments).

The online version of this article includes the following figure supplement(s) for figure 4:

**Figure supplement 1.** Median time-to-citation, per city, as a function of the average 10 year department-specific NIH funding.

and gerontology, hematology, rheumatology, and oncology) under the 'general and internal medicine' specialty label because a large portion of the funding for these specialties occurs through the NIH department combining name of 'internal medicine/medicine' (i.e. there was no specific department labels for these subset of specialties).

Average annual department-specific NIH funding correlated strongly with the absolute number of total citations, in the past decade, across all specialties (*Figures 4 and 5*): pathology (Spearman's rho = 0.73, p=1.2 × 10$^{-10}$), neurology (rho = 0.70, p<2.2 × 10$^{-16}$), pediatrics (rho = 0.67, p<2.2 × 10$^{-16}$), radiology (rho = 0.64, p=4.8 × 10$^{-12}$), internal medicine (rho = 0.60, p<2.2 × 10$^{-16}$), dermatology (rho = 0.57, p=1.2 × 10$^{-9}$), urology and nephrology (rho = 0.57, p=3.0 × 10$^{-13}$), emergency medicine (rho = 0.52, p=2.1 × 10$^{-7}$), anesthesiology (rho = 0.48, p=9.0 × 10$^{-6}$), and infectious diseases (rho = 0.41, p=0.006). *Figure 5* depicts city-labeled scatterplots that highlight, both, American cities (and institutions) that were successful at translating research back to the practice and cities that were particularly efficient. *Figure 4* illustrates the cumulative correlation of NIH funding, across all medical and surgical specialties. In sharp contrast, to both the department specific and cumulative funding associations with

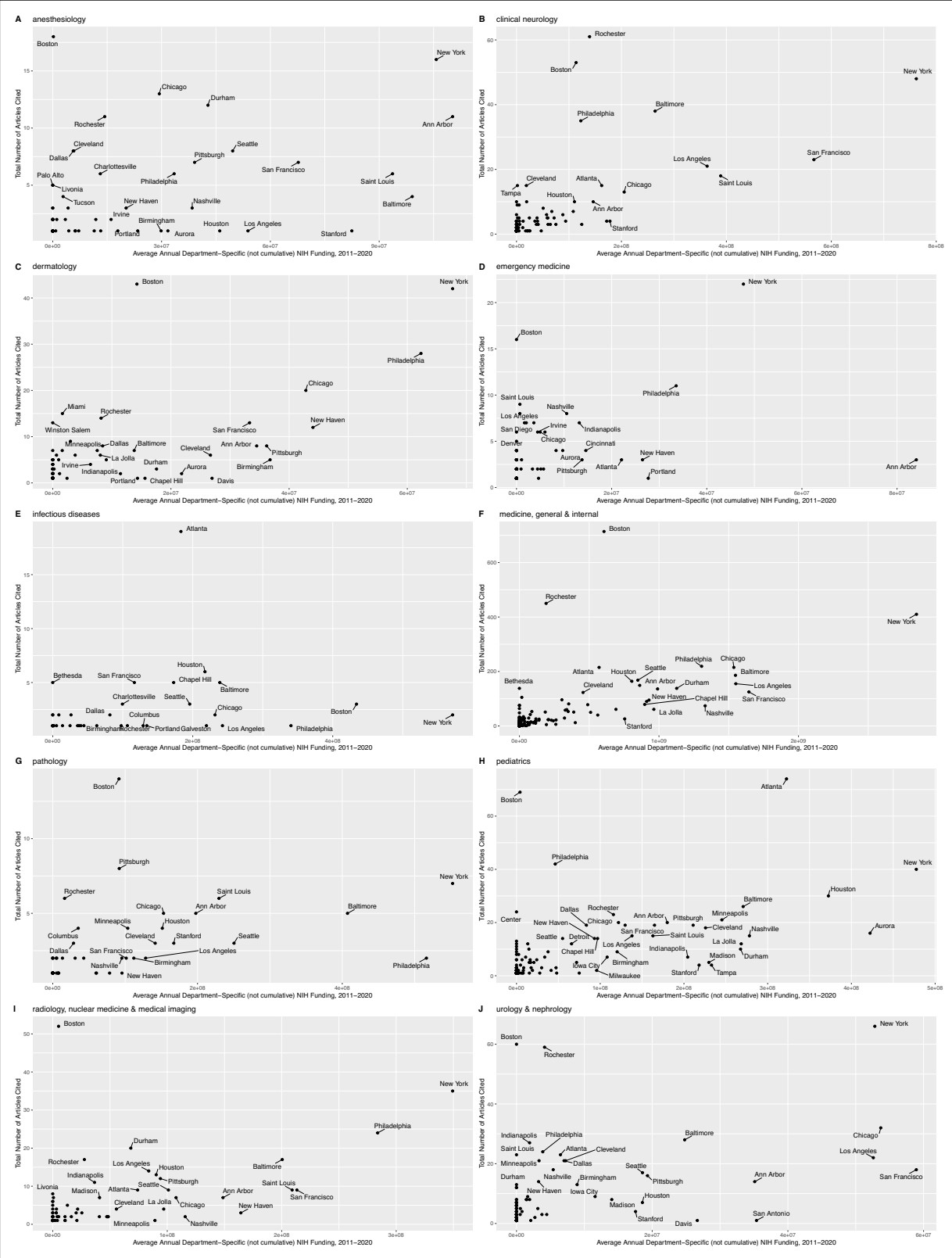

**Figure 5.** Number of articles cited, per city, as a function of the average 10 year department-specific NIH funding. Scatterplots of absolute number of articles cited, per city, as a function of the average 10 year department-specific NIH funding, for (**A**) anesthesiology, (**B**) clinical neurology, (**C**) dermatology, (**D**) emergency medicine, (**E**) infectious diseases, (**F**) medicine (general and internal), (**G**) pathology, (**H**) pediatrics, (**I**) 'radiology, nuclear medicine and medical imaging,' and (**J**) urology and nephrology. (**NB:**) The 'medicine, general and internal' plot, in panel **F**, combines the results for

*Figure 5 continued on next page*

*Figure 5 continued*

nine specialties (cardiac and cardiovascular systems, critical care medicine, endocrinology and metabolism, gastroenterology and hepatology, geriatrics and gerontology, hematology, rheumatology, and oncology – in addition to 'general and internal medicine') because a large portion of the funding for these eight specialties occurs through the NIH department combining name of 'internal medicine/medicine' (i.e. there was no specific department labels for these subset of specialties). It is also important to note that relevant funding may come from other NIH departments (e.g. 'cell biology'), and this is reflected in *Figure 4*.

The online version of this article includes the following figure supplement(s) for figure 5:

**Figure supplement 1.** Proportion of articles cited by city and country.

**Figure supplement 2.** Density maps illustrating the primary affiliation of articles in cited in UpToDate.

number of citations, NIH funding did not correlate with time-to-citation in any specialty (*Figure 4—figure supplement 1*).

Given the strength of correlation of NIH funding with the absolute number of citations across all medical specialties, we also sought to quantify the cost of one new added citation to the point-of-care (i.e. the slope) using a simple linear model. More concretely, we defined the model as a linear function between the average number of UpToDate citations from each city over the past 10 years and the average annual NIH department-specific funding between 2011 and 2020. This estimate may be interpreted as the approximate (indirect) cost of bringing clinical research to the bedside in NIH funding dollars, with the intercept being proportional to the initial investment 'set-up' cost. In descending order, a new citation at the point-of-care costs: $48,086.18 per new point-of-care UpToDate citation from urology and nephrology journals (± SE of $7410.68 and an intercept of $470,546.67), $34,529.29 dollars from dermatology journals (± $4043.66 and an intercept = $251,133.85), $13,286.72 from general and internal medicine specialty journals (± $746.58 and an intercept = $673,780.86), $10,655.93 dollars in emergency medicine (± $2795.21 and an intercept = $265,336.27), $6,832.46 dollars from pediatrics journals (± $756.08 and an intercept = $325,662.10), $6482.30 dollars from anesthesiology journals (± $1393.98 and intercept = $206,374.57), $6,227.91 dollars from radiology journals (± $1019.13 and intercept = $254,528.95), $6106.92 dollars from neurology journals (± $607.81 and intercept = $261,566.15), and $874.85 dollars from pathology journals (± $229.67 and an intercept = $174,163.00). The model was not significant for infectious diseases.

We subsequently generated US-focused maps to highlight, per specialty, institutions, and cities successful at translating clinical research from specialty journals to the bedside (*Figure 5—figure supplements 1 and 2*).

## Discussion

We have demonstrated that, depending on specialty, 0.4–2.4% of published clinical research is eventually cited in UpToDate. Our analysis also revealed several alarming trends: most clinical trials are never cited at the point-of-care – less than 1 in 10 phase III clinical trials are ever cited in 9 of 18 medical specialties. In the best-performing specialty (general and internal medicine), this citation reached a peak of 36%; that is, at least, 64% of trials are never cited. This was in line with a recent manual review of 125 randomized interventional clinical trials published in 2009–2010 in three disease areas (ischemic heart disease, diabetes mellitus, and lung cancer), which demonstrated only 26.4% of trials fulfilled four conditions of informativeness: importance of the clinical question, trial design, feasibility, and reporting of results (*Hutchinson et al., 2022*).

This trend was generally consistent among other higher quality-of-evidence research; 9 of 18 specialties had a citation rate of <13% for practice guidelines. Comparatively, while less than 1% of published case reports are ever cited, they represent one of the most commonly cited article types. For some specialties (e.g. dermatology), case-reports represented nearly a third of newly added references. The persistence of case reports as a resource to guide practice is itself not necessarily problematic; some are helpful in certain circumstances (e.g. to address rare conditions or phenomena that are hard to evaluate via other means). However, in some specialties, our results suggest that case reports outnumber most article types in UpToDate reference lists, including higher quality of evidence such as meta-analyses, systematic reviews, practice guidelines, and clinical trials. Further investigation of these case studies will highlight unmet clinical needs/questions that should be addressed with higher quality of evidence.

Reassuringly, a subset of specialties (e.g. 'cardiac and cardiovascular systems,' 'general and internal medicine,' and 'oncology') did incorporate higher quality of evidence into point-of-care reference lists, with more clinical trials cited than all other specialties cumulatively. In-depth investigation of differences between specialties in how clinical trials are designed/funded and how practice guidelines are formulated will likely reveal strategies to translating clinical research that should be applied more broadly.

Exploring over- and under-represented topics provided a fascinating perspective on how specialties prioritized particular topics, treatment paradigms, and clinical discoveries over the past decade. While a more thorough investigation is warranted, our preliminary study revealed that some specialties demonstrated a clear bias toward particular disease topics and treatment paradigms (e.g. cardiac and cardiovascular systems and oncology), while others were far more diverse (e.g. endocrinology and metabolism). The strong correlation of number of citations with NIH funding (both department specific and cumulatively) suggests that funding may, in part, dictate the research focus and, thus, which references are ultimately successful in making it back to the point-of-care.

## Limitations

There are many possible reasons for the low rate of citation of published research noted in our analysis (e.g. it is possible that some of the published research does not adequately answer a particular clinical question). It is also quite likely that the problem is at least partly one of translation. Both practice guidelines and clinical trials have a low-citation rate despite their design and implementation requiring uncertainty or equipoise surrounding two or more care options (i.e. they are designed to help clinicians choose one treatment or diagnostic approach over another). Thus, the low rate of citation rate of clinical trials, practice guidelines, and other high-quality evidence (e.g. systematic reviews) itself suggests a translational block. To address this limitation of various factors causing the low-citation rate, we explicitly investigate the rate of citation of various article types (as well as analyze topic distributions of these article types) separately and cumulatively to delineate the quality of evidence independently of the global citation rate.

Although it represents the largest and most comprehensive point-of-care resource, UpToDate is also just one perspective of how clinicians synthesize and integrate clinical research. Besides scholarly medical, nursing, and pharmacy journals (as major examples), many additional sources of information are readily available and accessed by these diverse stakeholders besides UpToDate. Examples include in person and online professional society communications and meetings, daily inter-professional interactions, local health system guideline consensus groups, access to clinicians who practice in 'centers of excellence,' and point-of-care decision support in electronic medical records. However, these are too informal and inaccessible for a systematic and comprehensive analysis of the translational highway between the clinical research enterprise and medical practice. Thus, UpToDate is a small but robust window into mapping and modeling translation.

We also recognize that the relevance of UpToDate varies by specialty and training status, and thus, its contents do not necessarily reflect the breadth or depth of medical care provided in a subset of medical specialties (and by extension, the body of evidence that underpins that care). Thus, while we use citation in UpToDate as a metric of translation, citation does not necessarily indicate actual changes in practice; rather citation represents adoption of knowledge to support current approaches, inform new changes in practice, or highlight points of controversy.

Importantly, strengthening our conclusions, UpToDate does serve as a reliable source of a specialty-based clinician-driven curation of the broader literature; its regularly updated reference lists accurately represent a clinician's perspective on the ever-expanding literature (*Shimizu et al., 2018*; *Maviglia et al., 2002*; *Lucas et al., 2004*; *Marshall et al., 2013*; *Bonis et al., 2008*). Thus, rather than viewing our analysis as a comprehensive look at all evidence that underpins all care, we suggest that this analysis be viewed as a standardized (cursory) survey of a fixed set of clinicians over the past decade on particular topics (defined by the scope of UpToDate articles).

Our division of the literature (and medical journals) into subspecialties using Clarivate's Journal Citation Reports admittedly does not capture the overlap/nuance of the boundaries between specialties (and journals); however, we believe it made our analyses much clearer and easier to understand. Where appropriate (e.g. citation rates of article types and cumulative NIH funding models), we

analyzed all specialties together to enable us to retain a bird's eye view on global trends across all 18 medical specialties.

## Conclusions

Tracing the path of clinical research into medical practice reveals substantial variation in how specialties prioritize and adopt clinical research into practice. The success of a subset of specialties in incorporating a larger proportion of published research, as well as high(er) quality of evidence, demonstrates the existence of translational strategies that should be applied more broadly. While the findings are largely descriptive and exploratory, the dataset and method described here are designed to generate hypotheses regarding the translation of biomedical research into practice. In designing the dataset, we sought to provide a baseline for monitoring the efficiency of research investments and ultimately lead to the development of mechanisms for weighing the efficacy of reforms to the biomedical scientific enterprise (e.g. quantifying impact at point-of-care rather than number of publications or citations).

# Materials and methods

We sampled all UpToDate articles (n=10,036 articles) multiple times over the past decade using the Internet Archive's WayBackMachine; capturing 169,203 unique versions over a median of 39 months per article (IQR 16–73 months). The WayBackMachine is a digital archive of the World Wide Web that preserves archived copies of defunct or revised web pages. The reference list of each UpToDate article was subsequently extracted a median of 14 times (IQR 6–25 times) over its respective sampling window. The reference lists were subsequently filtered to exclude non-research references as defined by MEDLINE. Our final dataset consisted of 83,423 unique references from 4055 journals newly cited in the sampling window. The first version of each UpToDate article served as a baseline to enable us to calculate the time-to-citation for all references (an additional brief Methods Supplement [Appendix 1] provides additional details about UpToDate) For clarity, throughout the text, we use the shorter phrase 'citation at the point-of-care' as equivalent to 'citation in an UpToDate article during our sampling window.' We subsequently filtered the references to those published in non-surgical specialties as defined by the Clarivate's Journal Citation Reports: 37,050 newly added unique references from 887 journals. To enable comparisons with the uncited literature, we used PubMed to identify all articles published during our sampling window in these 887 journals. These 2.4 million articles were similarly processed (i.e. matched to appropriate metadata).

We subsequently paired all references with the corresponding entries in PubMed to extract the associated abstracts, author affiliations, and date of publication. Thus, our final dataset for analysis represented a curated list of all references added over the past 10 years to UpToDate, alongside relevant metadata (such as journal, year of citation, author affiliations, etc.). We extracted the UMLS concepts from the paired abstracts using SciSpacy (*Neumann et al., 2019*), which enabled us to map the abstract free text to UMLS concepts (*Schuyler et al., 1993*). This is a similar pipeline used by PubMed to index articles for search engines and enabled us to extract 'high-level' concepts from the abstracts of all references. The performance of these algorithms (including validity and misclassification) are described elsewhere (*Schuyler et al., 1993*).

For this manuscript, we subsequently filtered the references to those published in non-surgical specialties as defined by the Clarivate's Journal Citation Reports (i.e. the categories specified in assessment of the impact factor): anesthesiology, cardiac and cardiovascular systems (i.e. cardiology), clinical neurology, critical care medicine, dermatology, emergency medicine, endocrinology and metabolism, gastroenterology and hepatology, geriatrics and gerontology, hematology, infectious diseases, medicine (general and internal), oncology, pathology, pediatrics, 'radiology, nuclear medicine and medical imaging' (i.e. radiology), rheumatology, and urology and nephrology. This filtered dataset subset included 37,050 newly added unique references from 887 journals, alongside relevant meta-data. To enable comparisons with the uncited literature, we used PubMed to identify all articles published during our sampling window in these 887 journals. These 2.4 million articles were similarly processed (i.e. matched to appropriate metadata).

For all analyses, summary statistics were generated using base functions in R v4.1. Where appropriate, p-values were corrected for multiple testing using Benjamini-Hochberg.

For all 887 journals, we also calculate two new indices: the CRI and the CII. Unlike the impact factor, these metrics exclusively quantify citations in point-of-care resources (i.e. UpToDate), rather than overall number of citations in other research publications and thus indirectly assess the presumed impact of any given journal on clinical practice. The CRI captures the long-standing impact of the journal over the past decade using a fraction of articles from the journal cited at least once in UpToDate and is defined as:

$$CRI_{decade} = \frac{Articles\ Cited\ in\ UpToDate\ in\ past\ decade}{Total\ Articles\ Ever\ Published\ past\ decade}.$$

Similarly, by using median time-to-citation, the CII captures journal specific trends in time-to-clinical-adoption (i.e. a measure of latency for each journal) that is distinct from the overall impact of the journal, defined as:

$$CII_{decade} = median\left(date\ of\ added\ citations\ in\ past\ decade - date\ of\ publication\right).$$

## Acknowledgements

Dr. Daniel B Jones for the critical review of the manuscript.

## Additional information

### Funding
No external funding was received for this work.

### Author contributions
Moustafa Abdalla, Conceptualization, Resources, Data curation, Software, Formal analysis, Supervision, Funding acquisition, Validation, Investigation, Visualization, Methodology, Writing – original draft, Project administration, Writing – review and editing; Salwa Abdalla, Data curation, Software, Formal analysis, Visualization, Methodology, Writing – review and editing; Mohamed Abdalla, Resources, Data curation, Software, Formal analysis, Supervision, Validation, Investigation, Methodology, Writing – review and editing

### Author ORCIDs
Moustafa Abdalla ⦿ http://orcid.org/0000-0002-2481-9753

### Decision letter and Author response
Decision letter https://doi.org/10.7554/eLife.82498.sa1
Author response https://doi.org/10.7554/eLife.82498.sa2

## Additional files

### Supplementary files
• Supplementary file 1. Tabulated summary of the clinical relevancy index (CRI) and clinical immediacy index (CII) for all journals, in all 18 medical specialties, rank-ordered by 5 year impact factor.

• MDAR checklist

### Data availability
We used the citation lists of all UpToDate articles published between 2011-2020. While all these citation lists are/were publicly available, we recognize the amount of work and effort required to collate and pre-process this data. As such, we have made publicly available the entire dataset used in this analysis to all readers at: https://www.8mlabs.org/uptodate/rawdataset.

The following dataset was generated:

| Author(s) | Year | Dataset title | Dataset URL | Database and Identifier |
|---|---|---|---|---|
| Abdalla M, Abdalla S, Abdalla M | 2023 | Raw Files | https://www.8mlabs.org/uptodate/rawdataset | 8mlabs, eLife_UpToDate_Abdalla2023 |

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

# Appendix 1

## Methods

### UpToDate details

There are more than 6300 physicians from 49 countries who serve as authors and/or editors on UpToDate. These physicians are experts in their fields and are often recruited by editors and/or may volunteer directly to contribute to UpToDate (at the discretion of the editors). Per UpToDate: 'the editorial team updates content continuously and is instructed to be transparent about the rigorous editorial process; all topics include the authors' and editors' names, specialties, and academic affiliations. In addition, UpToDate accepts no advertising or sponsorships.' Conflict of interest disclosures is available for all authors and editors.

## Results

### What topics are over-(or under-)represented in abstracts cited in point-of-care resources, compared to uncited literature?

Do these topics explain variation in time-to-citation? Abstract contents (as assessed using Unified Medical Language System concepts or terms) can significantly explain citation (versus not) in point-of-care resources, as well as variation in time-to-citation among cited abstracts (*Figure 3* and *Figure 3—figure supplements 1–4*). While results for all 18 specialties are fascinating and informative, here, we focus on two specialties.

### Cardiac and cardiovascular systems

In comparison to the 188,206 articles published in cardiology journals during our sampling window, the 2720 cited articles were significantly enriched for quantitative concepts (e.g. 'confidence interval,' 'person-years,' and 'median;' *Figure 3a*). Interestingly, 'HR' was substantially overrepresented compared to 'OR' (or 'relative risk'), suggesting a focus of the cited abstracts on 'death,' 'mortality,' and 'morbidity' in 'randomized trials' studies (terms also overrepresented in the cited abstracts). This was consistent with our observation that cardiology was one of six specialties that relied more heavily on higherquality of evidence, as discussed above. A subset of clinical concepts was also similarly enriched in cited abstracts (compared to uncited literature), including terms such as 'atrial fibrillation'/'arrhythmia,' '(hypertrophic) cardiomyopathy,' 'heart failure,' and 'syncope.' In sharp contrast, 'acute myocardial infarction,' 'ischemia,' 'plaque,' 'restenosis,' and 'coronary angiography' were significantly underrepresented in cited abstracts (*Figure 3—figure supplement 1*). When limited to the 2720 articles added to UpToDate, concepts such as 'venous thrombosis,' 'CHA$_2$DS$_2$-VASC score,' 'dual anti-platelet therapy,' 'pulmonary embolus,' and 'Everolimus-eluting stents' were associated with a shorter time-to-citation (*Figure 3—figure supplement 2*). Comparatively, articles focused on 'angina pectoris,' 'mitral valve prolapse,' 'cardiac catheterization,' 'transesophageal echocardiography,' and 'doppler imaging' had a longer time-to-citation (*Figure 3—figure supplements 3 and 4*).

### Endocrinology and metabolism

When compared to the 183,702 articles published in endocrinology journals, the 2181 cited abstracts were enriched for a diverse array of disease terms (e.g. 'Cushing syndrome,' 'polycystic ovary syndrome,' 'hypothyroidism,' 'type 1 diabetes,' and 'Graves disease') and associated complications (e.g. 'hypoglycemia,' 'fractures,' 'osteoporosis,' and 'quality of life;' *Figure 3b*). Quantitative terms were notably absent from cited abstracts. Unlike cardiology, there is no particular bias toward a subset of diseases in the cited literature. In fact, when exploring underrepresented words among cited articles (versus uncited references), the few clinical terms revolved around 'insulin' and 'adipose tissue.' All other terms underrepresented were reflective of more basic research work (e.g. 'gene expression,' 'mRNA,' 'in vitro,' 'animals,' and 'RT-PCR;' *Figure 3—figure supplement 1*). When limited to the 2181 references added to UpToDate, quantitative concepts (e.g. 'confidence interval') and clinical trial-related terms (e.g. 'participants,' 'interventions,' and 'pooled') were associated with a shorter time-to-citation (*Figure 3—figure supplement 2*), even if they were not independently or significantly predictive of whether an article would be cited. Adrenal and sex-related hormones (e.g. 'CRH,' 'dehydroepiandrosterone,' '17-hydroxyprogestrone,' and 'E2') were associated with a longer time-to-citation, as was 'Paget's disease,' 'premenopausal,' and some pregnancy-related concepts (e.g. 'placenta' and 'HCG'; *Figure 3—figure supplements 3 and 4*).

