## [Editor Report]

This paper presents an analysis of how clinical research makes its way into clinical practice in different medical specialties and across time. This is an important analysis since it has the potential to identify bottlenecks in the pathway from research to practice. The strength of evidence is convincing in that a substantial number of publications are considered over a wide time frame.

---

## [Decision Letter]

**Decision letter after peer review:**

Thank you for submitting your article "Tracing the Path of 37,050 Studies into Practice Across 18 Specialties of the 2.4 million Published between 2011-2020" for consideration by *eLife*. Your article has been reviewed by 2 peer reviewers, including Jennifer Flegg as Reviewing Editor and Reviewer #1, and the evaluation has been overseen by a Senior Editor. The following individual involved in the review of your submission has agreed to reveal their identity: Alan Schroeder (Reviewer #2).

Essential revisions:

1) Could something more than descriptive and exploratory be done? I feel like once the effect of one covariate is accounted for the significance of other variables may be changed. A multivariate statistical model could help address this.

2) Can possible solutions be proposed to the issues that were found?

3) The paper is long. The introduction (1.5 pages) could be shortened substantially. Where is the current knowledge gap lie and why does it need to be filled?

4) In the Introduction where you cite data on UpToDate (UTD) usage – where does this data come from?

5) The sampling strategy – this was truly done every 3 months since 2011? Specifically for this project?

6) The Methods section would benefit from a bit more background on UpToDate. Some of this information may be proprietary, but if not – how are authors selected? How do they deal with financial COI? Are the submissions peer-reviewed? What is the process/cadence for updating articles? Are there other methods for quality control? This information – in addition to an elaboration on the previously published literature assessing the impact of UTD on practice – would help to provide additional context for the findings. Given the current length of the paper, could consider adding this as a supplement.

7) There is a lot of information in this paper – it is very long and at times feels a bit scattered. Please consider simplifying the message.

*Reviewer #1 (Recommendations for the authors):*

In this work, the authors set out to trace the path from clinical research to clinical policy/practice. A major strength of this work is that a substantial number of publications are considered over a wide time frame. A weakness is that the statistical analysis is limited and could have been more sophisticated. I'm not convinced that the work will have a significant impact on the field since the findings are descriptive only.

1) Can anything more sophisticated be done for the statistical analysis rather than just correlations? I feel like once the effect of one covariate is accounted for the significance of other variables may be changed. A multivariate statistical model could help address this.

2) Could something more than descriptive and exploratory be done? Can possible solutions be proposed to the issues that were found?

*Reviewer #2 (Recommendations for the authors):*

I applaud the authors (who include a section editor for UTD) for trying to better understand the current landscape of the complicated link between evidence and practice. My main critique is that there is a lot of information in this paper – it is very long and at times feels a bit scattered. I think it would have been reasonable to split this into several papers, as there appear to be multiple aims (type of article, type of funding, field, content of abstract, etc.).

The other main critique/suggestion is that all of these analyses are based on UpToDate publications. UpToDate is effectively being used as a surrogate for practice (termed "point of care" in this paper) and this is arguably a big leap. While this issue is acknowledged in the Limitations section, I do think it could be flushed out more, as suggested below. For whatever it is worth, I can attest that in my own daily clinical experience in a large teaching hospital, UTD is indeed a main "go to" for many clinical questions. But that may simply represent the local culture.

Suggestions for improvement:

1. The paper is long. The introduction (1.5 pages) could be shortened substantially. Where is the current knowledge gap lie and why does it need to be filled?

2. In the Intro where you cite data on UTD usage – where does this data come from?

3. I am still trying to wrap my head around the sampling strategy. This was truly done every 3 months since 2011? Specifically for this project?

4. The Methods section would benefit from a bit more background on UpToDate. Some of this information may be proprietary, but if not – how are authors selected? How do they deal with financial COI? Are the submissions peer-reviewed? What is the process/cadence for updating articles? Are there other methods for quality control? This information – in addition to an elaboration on the previously published literature assessing the impact of UTD on practice – would help to provide additional context for the findings. Given the current length of the paper, could consider adding this as a supplement.

---

## [Author Response]

Essential revisions:1) Could something more than descriptive and exploratory be done? I feel like once the effect of one covariate is accounted for the significance of other variables may be changed. A multivariate statistical model could help address this.

We agree with the reviewer that the analysis was largely descriptive – we had two aims: (a) describe the collection of the dataset and the variables contained therein; and (b) provide a map of how clinical research makes its way into clinical practice in different medical specialties and across time to enable us to identify potential bottlenecks in the pathway from research to practice. Our focus on these goals is to leverage the strength of evidence available from looking at a substantial number of publications over a wide time frame. This is because a more complex multivariate analysis would require a firm understanding of specialty-specific research cycles and nuances to clinical adoptions. This research would best be placed in individual specialty-specific manuscripts and written with co-authors from the various specialties. The clinical authors of this manuscript are all in surgical subspecialities. The complete dataset, with extensive metadata as well as with complete pairing with PubMed and Clarivate databases, has now been made publicly available with no restrictions at: [https://www.8mlabs.org/uptodate/rawdataset]

Having said that, we added a new analysis that looks at the interaction of the fraction of open-source articles with the journal impact factor. Our hypothesis was that open-source articles/journals may influence time-to-citation or fraction of articles cited from a particular journal. Further, we did not believe there would be specialty-specific differences to this effect or specialty-specific factors that we would need to account for. Thus, this served as a straight-froward, but robust, multivariate analysis that extends our work beyond being simply descriptive. In future work, as aforementioned, we envision increasingly more complex analyses that spin off this dataset that we have created.

2) Can possible solutions be proposed to the issues that were found?

We have expanded several paragraphs in the discussion that proposes some solutions to the issues that were found. We hope our manuscript sparks additional discussion about how the various specialties can learn from pragmatic best practices across, the sometimes artificial, specialty boundaries.

3) The paper is long. The introduction (1.5 pages) could be shortened substantially. Where is the current knowledge gap lie and why does it need to be filled?

The introduction was revised and shortened to 1 page, primarily focusing on defining the current knowledge gap (understanding the factors that influence the translation of research to the point-of-care) and why it needs to be filled (means to generating solutions to hurdles to the clinical adoption of research).

4) In the Introduction where you cite data on UpToDate (UTD) usage – where does this data come from?

This data comes from UpToDate and the citations have been updated to reflect this.

5) The sampling strategy – this was truly done every 3 months since 2011? Specifically for this project?

The sampling was done through the Internet Archive’s WayBackMachine, which is a digital archive that preserves archived copies of defunct or revised web pages. Our sampling of this resource was done specifically for this project, but WayBackMachine archives many webpages. The Methods section was amended to clarify this.

6) The Methods section would benefit from a bit more background on UpToDate. Some of this information may be proprietary, but if not – how are authors selected? How do they deal with financial COI? Are the submissions peer-reviewed? What is the process/cadence for updating articles? Are there other methods for quality control? This information – in addition to an elaboration on the previously published literature assessing the impact of UTD on practice – would help to provide additional context for the findings. Given the current length of the paper, could consider adding this as a supplement.

We have added a Supplement that addresses the aforementioned questions.

7) There is a lot of information in this paper – it is very long and at times feels a bit scattered. Please consider simplifying the message.

We have shortened the introduction, revised the methods, and moved a portion of the analyses to supplement. Additional analyses requested during this review process were also added as supplements to ensure simplicity of message: “we have curated a powerful new dataset that will allow us to quantify and understand the modifiable factors that influence the clinical adoption of research.” The complete dataset, with extensive metadata as well as with complete pairing with PubMed and Clarivate databases, has now been made publicly available with no restrictions at: [https://www.8mlabs.org/uptodate/rawdataset]

Reviewer #1 (Recommendations for the authors):In this work, the authors set out to trace the path from clinical research to clinical policy/practice. A major strength of this work is that a substantial number of publications are considered over a wide time frame. A weakness is that the statistical analysis is limited and could have been more sophisticated. I'm not convinced that the work will have a significant impact on the field since the findings are descriptive only.1) Can anything more sophisticated be done for the statistical analysis rather than just correlations? I feel like once the effect of one covariate is accounted for the significance of other variables may be changed. A multivariate statistical model could help address this.

We agree with the reviewer that the analysis was largely descriptive – we had two aims: (a) describe the collection of the dataset and the variables contained therein; and (b) provide a map of how clinical research makes its way into clinical practice in different medical specialties and across time to enable us to identify potential bottlenecks in the pathway from research to practice. Our focus on these goals is to leverage the strength of evidence available from looking at a substantial number of publications over a wide time frame. This is because a more complex multivariate analysis would require a firm understanding of specialty-specific research cycles and nuances to clinical adoptions. This research would best be placed in individual specialty-specific manuscripts and written with co-authors from the various specialties. The clinical authors of this manuscript are all in surgical subspecialities. The complete dataset, with extensive metadata as well as with complete pairing with PubMed and Clarivate databases, has now been made publicly available with no restrictions at: [https://www.8mlabs.org/uptodate/rawdataset]

Having said that, we added a new analysis that looks at the interaction of the fraction of open-source articles with the journal impact factor. Our hypothesis was that open-source articles/journals may influence time-to-citation or fraction of articles cited from a particular journal. Further, we did not believe there would be specialty-specific differences to this effect or specialty-specific factors that we would need to account for. Thus, this served as a straight-froward, but robust, multivariate analysis that extends our work beyond being simply descriptive. In future work, as aforementioned, we envision increasingly more complex analyses that spin off this dataset that we have created.

2) Could something more than descriptive and exploratory be done? Can possible solutions be proposed to the issues that were found?

We have expanded several paragraphs in the discussion that proposes some solutions to the issues that were found. We hope our manuscript sparks additional discussion about how the various specialties can learn from best practices across, the sometimes artificial, specialty boundaries.

Reviewer #2 (Recommendations for the authors):I applaud the authors (who include a section editor for UTD) for trying to better understand the current landscape of the complicated link between evidence and practice. My main critique is that there is a lot of information in this paper – it is very long and at times feels a bit scattered. I think it would have been reasonable to split this into several papers, as there appear to be multiple aims (type of article, type of funding, field, content of abstract, etc.).The other main critique/suggestion is that all of these analyses are based on UpToDate publications. UpToDate is effectively being used as a surrogate for practice (termed "point of care" in this paper) and this is arguably a big leap. While this issue is acknowledged in the Limitations section, I do think it could be flushed out more, as suggested below. For whatever it is worth, I can attest that in my own daily clinical experience in a large teaching hospital, UTD is indeed a main "go to" for many clinical questions. But that may simply represent the local culture.Suggestions for improvement:1. The paper is long. The introduction (1.5 pages) could be shortened substantially. Where is the current knowledge gap lie and why does it need to be filled?

The introduction was revised and shortened to 1 page, primarily focusing on defining the current knowledge gap (understanding the factors that influence the translation of research to the point-of-care) and why it needs to be filled (means to generating solutions to hurdles to the clinical adoption of research).

2. In the Intro where you cite data on UTD usage – where does this data come from?

This data comes from UpToDate and the citations have been updated to reflect this.

3. I am still trying to wrap my head around the sampling strategy. This was truly done every 3 months since 2011? Specifically for this project?

The sampling was done through the Internet Archive’s WayBackMachine, which is a digital archive that preserves archived copies of defunct or revised web pages. Our sampling of this resource was done specifically for this project, but WayBackMachine archives many webpages. The Methods section was amended to clarify this.

4. The Methods section would benefit from a bit more background on UpToDate. Some of this information may be proprietary, but if not – how are authors selected? How do they deal with financial COI? Are the submissions peer-reviewed? What is the process/cadence for updating articles? Are there other methods for quality control? This information – in addition to an elaboration on the previously published literature assessing the impact of UTD on practice – would help to provide additional context for the findings. Given the current length of the paper, could consider adding this as a supplement.

We have added a Supplement that addresses the aforementioned questions.